# Faecal microbiota transplantation for the treatment of diarrhoea induced by tyrosine-kinase inhibitors in patients with metastatic renal cell carcinoma

Gianluca Ianiro [1,7], Ernesto Rossi [2,7], Andrew M. Thomas [3], Giovanni Schinzari[2], Luca Masucci [4], Gianluca Quaranta[4], Carlo Romano Settanni[1], Loris Riccardo Lopetuso[1], Federica Armanini[3], Aitor Blanco-Miguez[3], Francesco Asnicar[3], Clarissa Consolandi[5], Roberto Iacovelli [2], Maurizio Sanguinetti [4], Giampaolo Tortora[2], Antonio Gasbarrini[1], Nicola Segata [3,6,7] & Giovanni Cammarota [1,7 ✉]

Diarrhoea is one of the most burdensome and common adverse events of chemotherapeutics, and has no standardised therapy to date. Increasing evidence suggests that the gut microbiome can influence the development of chemotherapy-induced diarrhoea. Here we report findings from a randomised clinical trial of faecal microbiota transplantation (FMT) to treat diarrhoea induced by tyrosine kinase inhibitors (TKI) in patients with metastatic renal cell carcinoma (ClinicalTrials.gov number: NCT04040712). The primary outcome is the resolution of diarrhoea four weeks after the end of treatments. Twenty patients are randomised to receive FMT from healthy donors or placebo FMT (vehicle only). Donor FMT is more effective than placebo FMT in treating TKI-induced diarrhoea, and a successful engraftment is observed in subjects receiving donor faeces. No serious adverse events are observed in both treatment arms. The trial meets pre-specified endpoints. Our findings suggest that the therapeutic manipulation of gut microbiota may become a promising treatment option to manage TKI-dependent diarrhoea.

[1] Digestive Disease Center, Fondazione Policlinico Universitario "A. Gemelli" IRCCS, Università Cattolica del Sacro Cuore, Largo A. Gemelli 8, 00168 Rome, Italy. [2] Medical Oncology Unit, Comprehensive Cancer Center, Fondazione Policlinico Universitario "A. Gemelli" IRCCS, Università Cattolica del Sacro Cuore, Largo A. Gemelli 8, 00168 Rome, Italy. [3] Department of Cellular, Computational and Integrative Biology (CIBIO), University of Trento, Via Sommarive 9, 38123 Povo, Trento, Italy. [4] Microbiology Unit, Fondazione Policlinico Universitario "A. Gemelli" IRCCS, Università Cattolica del Sacro Cuore, Largo A. Gemelli 8, 00168 Rome, Italy. [5] Institute of Biomedical Technologies (IBT), Italian National Research Council (CNR), Via Fratelli Cervi, 93, 20090 Segrate, Milan, Italy. [6] IEO, European Institute of Oncology IRCCS, Via Adamello 16, 20139 Milan, Italy. [7] These authors contributed equally: Gianluca Ianiro, Ernesto Rossi, Nicola Segata, Giovanni Cammarota. ✉email: giovanni.cammarota@unicatt.it

Renal cell carcinoma (RCC) remains one of the most burdensome urological cancers, as its incidence is increasing worldwide[1], and a considerable proportion of patients present with metastatic disease at diagnosis[2]. The oral multi-targeted receptor tyrosine-kinase inhibitors (TKIs) sunitinib and pazopanib have dramatically improved clinical outcomes of patients with metastatic RCC (mRCC)[3], and are among the first-line options for this condition[4]. However, the efficacy of these drugs is commonly encumbered by diarrhoea, which occurs in nearly 50% of patients[5–7] and often requires dose reduction and temporary drug interruption[8]. To date, there are no standardised strategies for TKIs-related diarrhoea, and current recommendations are supported by limited evidence or real-life experience[9].

Increasing evidence suggests that gut microbiome could influence the development of TKIs-induced diarrhoea, as chemotherapy is known to alter the gut microbiome[10] and distinct microbial profiles have been found in patients with TKIs-induced diarrhoea[11]. In principle, the therapeutic modulation of gut microbiota could alleviate TKI-induced diarrhoea, and probiotics have been suggested as a possible treatment option for this condition, but little evidence supports this indication[12,13]. Faecal microbiota transplantation (FMT) is the infusion of healthy donor feces in the recipient's gut to cure a specific disease. FMT is clearly recognised as a highly effective treatment against recurrent *Clostridioides difficile* infection[14,15] and has shown promising results in other dysbiosis-associated disorders, including ulcerative colitis[16] or metabolic syndrome[17]. FMT has recently shown potential in treating refractory immune checkpoint inhibitor-associated colitis in two patients[18], but has never been investigated so far in chemotherapy-induced diarrhoea, although this condition is much more common than immunotherapy-dependent diarrhoea[5–7,19]. Here we report findings from a randomised clinical trial of donor FMT versus placebo FMT to treat diarrhoea induced by TKI in patients with metastatic renal cell carcinoma (NCT04040712). Donor FMT is more effective than placebo FMT in treating TKI-induced diarrhoea, and a successful engraftment is observed in subjects receiving donor faeces. Our findings suggest that the therapeutic manipulation of gut microbiota may become a treatment option to manage TKI-dependent diarrhoea.

## Results

Here we report clinical and microbiological results from a randomised clinical trial (NCT04040712) of FMT to treat TKI-induced diarrhoea in patients with mRCC. The primary outcome was the resolution of diarrhoea 4 weeks after the end of treatments. From August 2019 to December 2019 we enrolled 20 mRCC patients (F = 5, M = 15, mean age: 65 years) under treatment with pazopanib (16 subjects) or sunitinib (four subjects) without other prior medical conditions or recent antibiotic use (see online Methods) who were randomly assigned to receive FMT from healthy donors (D-FMT, 10 subjects, F = 2, M = 8, mean age: 63 years) or placebo (i.e. vehicle only) FMT (P-FMT, 10 subjects, F = 3, M = 7, mean age: 66 years) (Fig. 1a). Nineteen patients presented with a G2 diarrhoea, and one (in the D-FMT group) with a G3 diarrhoea, according to the *Common Terminology Criteria (CTC) for Adverse Events* (AE) version 4.0[20]. No significant differences in demographic and clinical characteristics were observed between the two groups at baseline (Supplementary Table 1). At the time of study enrolment, one patient in D-FMT group was on treatment with fentanyl and one patient in P-FMT group was on treatment with oxycodone. None of the patients started opiates during the study. We recruited two healthy donors for D-FMT (donor A and donor B, who donated to four and six patients, respectively) to minimise the donor effect[21]. FMT was performed only once for each patient, with an average of 57 g of feces per each procedure (range: 50–80 g).

We observed full resolution of diarrhoea 4 weeks after treatments in seven of the D-FMT group patients against none in the P-FMT group (primary outcome, 70% vs 0%, Fisher's exact test p = 0.003, Fig. 1b). One week after treatment, all D-FMT patients were diarrhoea-free, against three of the 10 control subjects (100% vs 30%, p = 0.02, after Bonferroni adjustment for multiple secondary outcomes), and only one D-FMT patient experienced recurrence of diarrhoea, versus all patients in the P-FMT group, within 2 weeks (90% vs 0% resolution, p = 0.0007). The effect of FMT on diarrhoea resolution became less pronounced at 8 weeks (30% D-FMT vs 0% P-FMT, p = 1), consistently with the single FMT treatment approach adopted in our study design. Nonetheless, D-FMT was significantly more effective than P-FMT in decreasing diarrhoea to G1 grade or lower not only after 1 week (100% vs 30%, p = 0.02), 2 weeks (100% vs 20%, p = 0.005), and 4 weeks of follow-up (100% vs 20%, p = 0.005), but also at 8 weeks of follow-up (80% vs 10%, p = 0.04, Fig. 1b). This finding suggests that there could be a long-lasting effect of even a single FMT infusion in mRCC patients. Importantly, TKI dosage had to be reduced due to the diarrhoea in three P-FMT patients based on the physician's decision, while no adjustment of TKI was necessary in the D-FMT group. No patients in the D-FMT group and six patients in the P-FMT group took loperamide to alleviate diarrhoea. TKI had to be temporarily stopped in four patients in P-FMT group (in three patients despite loperamide), while it was not interrupted in D-FMT group. Six patients in D-FMT group experienced transient constipation after FMT. No serious adverse events nor procedure-related complications were observed in both groups.

We then investigated the effects of FMT on the gut microbiome of D-FMT patients after 1, 2 and 4 weeks of follow-up (see Methods), by using shotgun metagenomic sequencing[22]. The composition of the gut microbiome from D-FMT and P-FMT groups did not show significant differences before treatment (p = 0.7, ADONIS), as expected from the random patient-to-treatment assignment. For all D-FMT patients, the post-FMT gut microbiome resembled more the composition of the gut microbiome of the donor compared to that of the pre P-FMT patients used as controls (Fig. 2a–c). This observation was confirmed when looking at the overall quantitative microbiome composition (Fig. 2a) as well as the fraction of shared species (Fig. 2b) and of shared strains (Fig. 2c), and remained significant (p < 0.05) for all three measures at 4 weeks post D-FMT. One week and 1 month after D-FMT we found, on average, more strains in the gut microbiome of the patients that we tracked from the donors compared to those present also in the pre-D-FMT samples, suggesting a very effective microbial engraftment rate after FMT. The 779 total pairwise strain comparisons we were able to analyse by StrainPhlAn analysis[23] spanned 53 species and 5 phyla (*Proteobacteria*, *Actinobacteria*, *Firmicutes*, *Bacteroidetes* and *Verrucomicrobia*). We found 232 strains present in more than one sample of which 53% were shared within D-FMT patients across different time points, 28% were shared between D-FMT patients and their respective donors, and 8.6% were shared between D-FMT samples of the same donor. We observed that D-FMT and donor samples shared significantly more strains between each other across all post-FMT time points than between P-FMT and D-FMT samples, with the highest effect observed just 1 week after FMT infusion (Fig. 2c, Supplementary Fig. 1). The most commonly shared species' strains between D-FMT patients and donors that points at successful strain engraftment, included the beneficial commensal *Akkermansia muciniphila* (Supplementary Fig. 2, <1% average abundance in patients, 10.9% abundance in

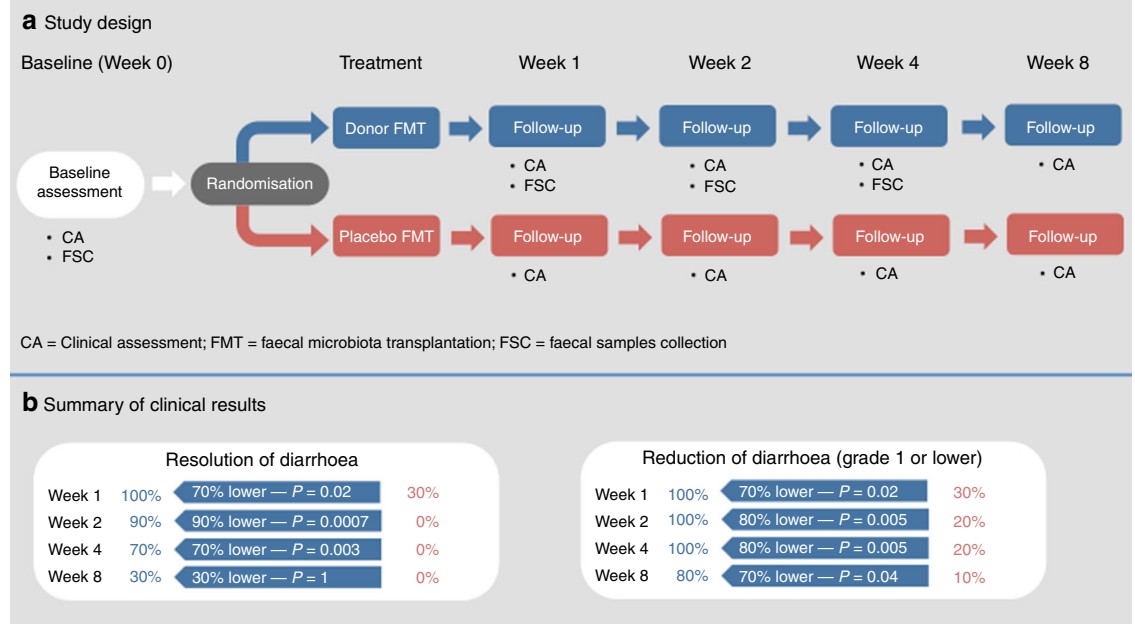

**Fig. 1 Study design and summary of clinical results. a** Study design: after baseline clinical and microbiome assessments, patients were randomised to receive either donor faecal microbiota transplantation (D-FMT) or placebo FMT (P-FMT). Then, all patients were followed up to 8 weeks (follow-up visits at 1, 2, 4 and 8 weeks after treatments). **b** Summary of clinical results: D-FMT was significantly more effective than P-FMT in achieving the complete resolution of TKI-induced diarrhoea at 1 week ($n = 10$ vs $n = 3$, 100% vs 30%, $p = 0.02$), 2 weeks ($n = 9$ vs $n = 0$, 90% vs 0%, $p = 0.0007$), 4 weeks (primary outcome, $n = 7$ vs $n = 0$, 70% vs 0%, $p = 0.003$), but not at 8 weeks of follow-up ($n = 3$ vs $n = 0$, 30% vs 0%, $p = 1$), consistently with the single FMT infusion approach of the study. Nonetheless, the benefit of D-FMT over P-FMT in decreasing diarrhoea up to G1 grade or lower remained significant for the whole follow-up period, including 1 week ($n = 10$ vs $n = 3$, 100% vs 30%, $p = 0.02$), 2 weeks ($n = 10$ vs $n = 2$, 100% vs 20%, $p = 0.005$), 4 weeks ($n = 10$ vs $n = 2$, 100% vs 20%, $p = 0.005$) and 8 weeks ($n = 8$ vs $n = 1$, 80% vs 10%, $p = 0.04$) after the end of treatments. Differences in cure percentages were determined with Fisher's exact test (with two-tailed $p$ values). The Bonferroni adjustment was applied for multiple secondary outcomes.

donor B) and several species in the *Bacteroides* phylum such as *Alistipes putredinis* (Fig. 2f, 2.0% average abundance in patients, 8.2% and 8.7% abundance in donor A and B, respectively) and *Barnesiella intestinihominis* (1.3% average abundance in patients, 4.1% abundance in donor B). On the other hand, species' strains shared within D-FMT patients between pre-FMT samples and different time points were mainly from the *Firmicutes* phylum and included *Roseburia inulinovorans* and *Faecalibacterium prausnitzii* (Fig. 2d, e). Overall, the clinical success of D-FMT was supported by strong donor strain engraftment. The species responsible for most donor-recipient strain transmissions may have a role in the therapeutic effect of D-FMT on TKI-dependent diarrhoea.

## Discussion

Altogether, in this study we found a clear and statistically supported benefit of FMT in improving TKI-dependent diarrhoea in patients with mRCC. Our clinical results match with microbiological findings, as we observed a successful engraftment after D-FMT at the strain level, suggesting that the beneficial effects of FMT may not only lay in changes to gut microbial community structure or membership, but also in strain-level diversity. However, our data are too preliminary to disentangle clearly the role of engraftment in driving clinical success. Also, we acknowledge that potential limitations of our study, including the use of water instead of patients' autologous faeces as comparator, or the absence of microbiome assessment after placebo FMT, could prevent us from drawing definitive conclusions on our findings. Although additional studies are needed to further consolidate these results, this study shows that FMT can improve clinically TKI-induced diarrhoea and favourably modulate the gut

microbiome of these patients. More widely, our findings suggest that the therapeutic manipulation of gut microbiota may be a fundamental step in the management of chemotherapy-dependent diarrhoea.

## Methods

**Study design.** This is a single-centre placebo-controlled, double-blind randomised clinical trial of donor FMT vs placebo FMT in patients with TKI-induced diarrhoea carried out in the Policlinico Universitario "A. Gemelli" IRCCS, an academic tertiary care centre based in Rome, Italy. The study was approved by the institutional review board/local ethics committee, and has been registered in *ClinicalTrials.gov* (Identifier: NCT04040712). Most patients were pre-evaluated within clinical practice before the official start of enrolment. All enroled subjects gave their written informed consent to participate in the study. This study was conducted by following the CONSORT guidelines (Supplementary Fig. 3, Supplementary Table 2)[24]. The full study protocol is available as Supplementary Note 1 in the Supplementary Information file.

**Study population.** Eligible subjects were 18 years old or older, already on treatment with TKIs (pazopanib or sunitinib) for metastatic RCC diagnosed by histology and measurable according to RECIST criteria version 1.1[25], who had developed diarrhoea of grade 2 or 3 according to the *Common Terminology Criteria (CTC) for Adverse Events* (AE) version 4.0[20] induced by these drugs. Patients must also have performed a CT scan no earlier than 4 weeks before enrolment, and present with: a good or intermediate prognostic assessment (according to criteria of the prognostic system of the International Metastatic RCC Database Consortium[26]); a performance status equal or lower than 2[27]; blood count, hepatic and kidney testing within normal limit. Finally, patients must have been able to give their consent to be included in the study.

Patients were excluded if they had another known cause of diarrhoea (e.g. infectious gastroenteritis, *Clostridium difficile* infection, coeliac disease, inflammatory bowel disease, irritable bowel syndrome, chronic pancreatitis and/or biliary salt diarrhoea), previous colorectal surgery or cutaneous stoma, food allergies, recent (<6 weeks) therapy with drugs that could possibly alter gut microbiota (e.g. antibiotics, probiotics, proton pump inhibitors, immunosuppressants and/or metformin), another cancer (except for surgically treated basocellular carcinoma), brain metastases, decompensated heart failure or

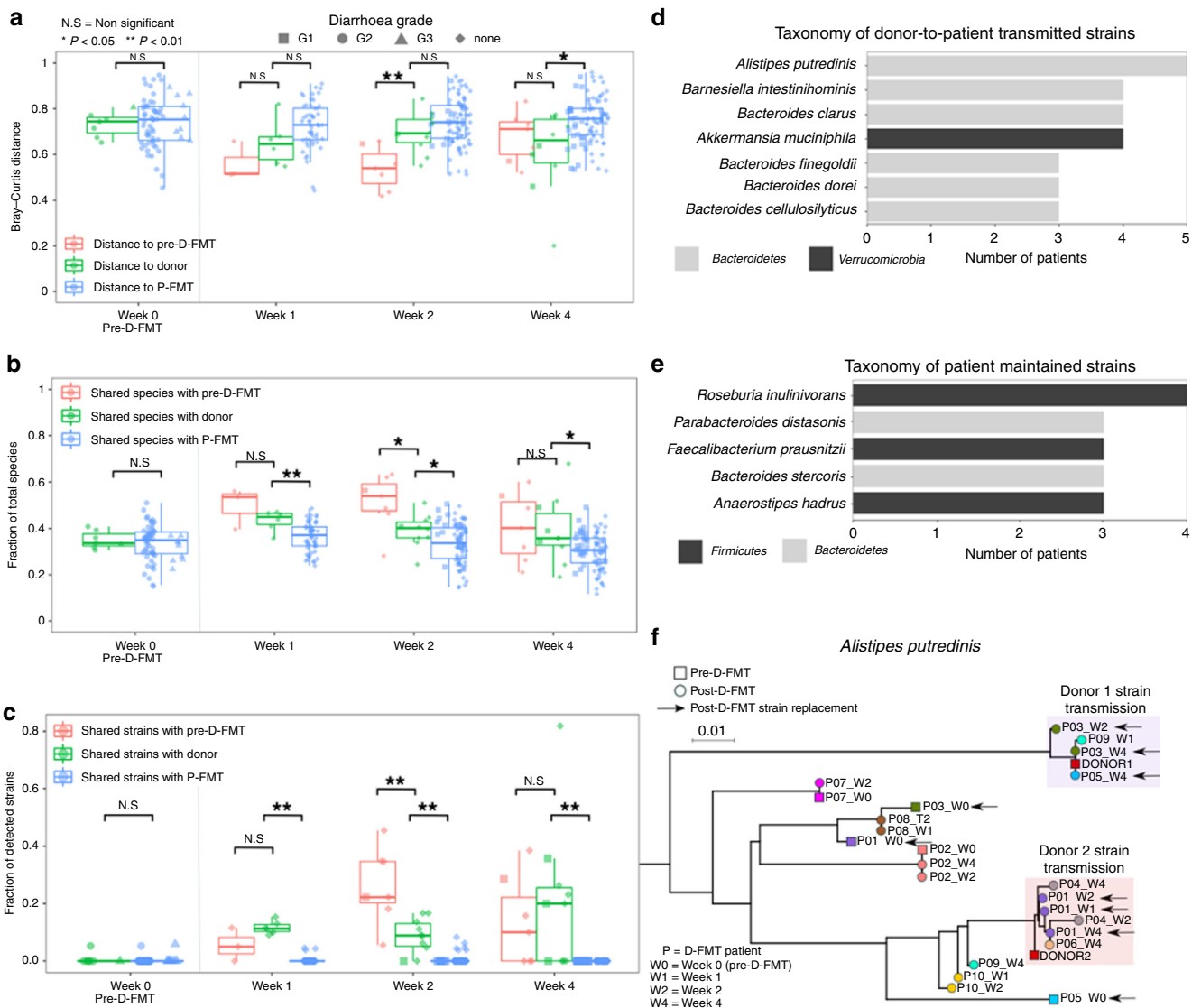

**Fig. 2 Long-term donor microbial strain engraftment in mRCC patients receiving D-FMT.** We evaluated the effect of D-FMT in patients by assessing how much the overall microbiome got closer to that of the donor after FMT using **a** pairwise Bray-Curtis distances calculated on species-level abundances (p values from left to right: $p = 0.74$, $p = 0.17$, $p = 0.06$, $p = 0.004$, $p = 0.177$, $p = 0.74$ and $p = 0.013$), **b** the fraction of shared species (p values from left to right: $p = 0.724$, $p = 0.381$, $p = 0.007$, $p = 0.018$, $p = 0.047$, $p = 0.889$ and $p = 0.04$), and **c** strains between pairs of samples (see Methods) (p values from left to right: $p = 0.18$, $p = 0.262$, $p = 1.5 \times 10^{-10}$, $p = 0.007$, $p = 2.8 \times 10^{-11}$, $p = 0.723$ and $p = 5.36 \times 10^{-14}$). All p values were calculated using a two-sided Wilcoxon rank-sum test. While all patients had a microbiome as distinct from the donors as control patients, after D-FMT the donor-recipient microbiome distance was reduced with significantly more species and strains shared. The donor's strains remained in the recipient for up to 30 days, highlighting the high engraftment potential of a single D-FMT treatment in the patients. Box-plots report median (central lines), 25th and 75th percentiles (box limits), and the upper and lower whiskers extend from the hinges to the largest (or smallest) value no further than ×1.5 interquartile range from the hinge, defined as the distance between the 25th and 75th percentiles. **d** Species with the highest number of D-FMT patients showing evidence of acquisition of the donor's strain and of **e** maintenance of pre-FMT strains. **f** Representative phylogenetic tree of *Alistipes putredinis* showing strain transmission and replacement events for this species between D-FMT patients and their respective donors.

heart disease with ejection fraction lower than 30%, severe respiratory insufficiency, psychiatric disorders, were pregnant or were unable to give informed consent.

Potentially eligible patients, based on these criteria, underwent the following exams to exclude other causes of diarrhoea: *C. difficile* (culture and toxin), bacterial culture for enteric pathogens, including *Salmonella, Shigella, Campylobacter, Escherichia coli O157 H7, Yersinia*, VRE (vancomycin-resistant Enterococci), MRSA (methicillin-resistant *Staphylococcus aureus*), Gram-negative MDR (multi-drug-resistant) bacteria, *Vibrio cholerae, Listeria monocytogenes*, Norovirus, protozoa and helminths/ova and parasites, faecal pancreatic elastase, C-reactive protein, erythrocyte-sedimentation rate, transglutaminase antibodies, total IgA and IgE and ileocolonoscopy.

All subjects who met inclusion criteria and tested negative for these exams were finally enrolled in the study. The first patient was enroled in August 2019, and the last patient was enroled in December 2019.

**Baseline assessments.** Before randomisation, demographic informations were collected and patients were evaluated for the severity of diarrhoea according to the National Cancer Institute Common Toxicity Criteria; NCI CTC version 4.0 (grade 0 = none; grade 1 = increase of <4 stools/day over pre-treatment; grade 2 = increase of 4–6 stools/day, or nocturnal stools; grade 3 = increase of ≥7 stools/day or incontinence or need for parenteral support for dehydration; grade 4 = physiologic consequences requiring intensive care, or hemodynamic collapse)[20].

In addition, patients were requested to give stool samples that were collected in a sterile, sealed container and stored at −80 °C for metagenomic assessment of gut microbiota.

**Study treatments.** After baseline assessments, patients were randomly assigned to donor FMT (D-FMT) or to placebo FMT (P-FMT). Patients in both groups

underwent a single FMT procedure. Both donor and placebo infusates were delivered by colonoscopy. Each patient in the donor FMT group received stool from one single donor. Placebo FMT consisted of 250-mL water. Loperamide was allowed as rescue anti-diarrhoeal medication if diarrhoea did not respond to experimental treatments.

**Selection of stool donors**. The selection of stool donors was performed by physicians from the Digestive Disease Unit (G.I., C.R.S. and G.I.) following protocols previously recommended by international guidelines, including: a questionnaire to address donor medical history; blood and stool exams to exclude potentially transmittable diseases; and a further questionnaire administered to selected donors the day of the feces collection to rule out any issue happened within the screening period[15]. Details of the donor screening protocol are available in the study protocol.

The assignment of faecal infusates from healthy donors to patients was done randomly, without any specific recipient–donor match, as suggested by international guidelines[15]. Two healthy subjects were selected as stool donors. At the time of their first donation, a sample of feces was collected and stored at −80 °C for microbiome analysis.

**Manufacturing of faecal infusate**. All faecal infusate samples were manufactured in the microbiology laboratory of our hospital, using fresh feces only. At least 50 g of feces, diluted in 250 mL of saline, were used for a single sample. We followed manufacturing protocols recommended by international guidelines for fresh feces[15]. Details of the manufacturing process are available in the study protocol.

**FMT procedures**. All procedures were performed by colonoscopy, under sedation. Patients in both groups underwent bowel cleansing with 4 L of macrogol (SELG ESSE) the day before the procedure. All procedures were performed by two expert endoscopists (L.R.L. and G.C.), using paediatric colonoscopes and carbon dioxide insufflation. Both faecal infusates and placebo infusates were delivered through the operative channel of the scope after reaching the more proximal point of the large bowel, using 50-mL syringes filled with the infusate during colonoscopy. The faecal infusate was delivered within 6 h after donor supply. After the procedures, patients were monitored in the recovery room of the endoscopy centre for nearly 3 h.

**Follow-up**. Follow-up visits were performed by physicians from the oncology unit (E.R., G.S., R.I. and G.T.). All patients were followed up for 3 months after the end of treatments. Follow-up visits were scheduled at week 1, week 2, week 4 and week 8, after the end of treatments, respectively. At each visit the following assessments were performed: (1) patients were evaluated for severity of diarrhoea following the National Cancer Institute Common Toxicity Criteria (NCI CTC) version 4.0[20]; (2) concomitant medications were recorded, and the use of loperamide was registered; (3) patients provided stool samples that were stored at −80 °C for microbiota analysis; (4) adverse events were recorded. Unscheduled follow-up visits were offered if requested by the patients.

**Microbiome analysis**. We collected and analysed via shotgun metagenomic sequencing a total of 45 stool samples encompassing four different time points of D-FMT patients ($n = 10$), one time-point of P-FMT patients ($n = 10$), and the two donors. Samples were collected in a sterile, airtight container for microbiome assessment, brought by patients to the hospital in a refrigerated bag, and put in a −80 °C freezer, within 12 h from the defecation event. DNA was extracted using the Danagene Microbiome Fecal DNA kit and sequenced on the Illumina NovaSeq platform with 150 × 2 chemistry. Metagenomic shotgun sequences were quality filtered using trim_galore discarding all reads of quality <20 and shorter than 75 nucleotides. Filtered reads were then aligned to the human genome (hg19) and the PhiX genome for human and contaminant DNA removal using bowtie2[28], yielding an average of 5.1 G bases in high-quality reads in each sample. Species-level profiles were calculated in bioBakery[29] using MetaPhlAn[30] version 3.0 (https://github.com/biobakery/MetaPhlAn). We performed strain-level analysis using StrainPhlAn[23] version 3.0, and same strains were defined using a cutoff of 0.01 normalised genetic distance calculated using the branch length of the strains' phylogenetic tree. The default parameters were used for StrainPhlAn which have been set to maximise a confident retrieval of strain-level profiles; nonetheless, not all the species/sample combinations identified by MetaPhlAn could be profiled at the strain level by StrainPhlAn and thus it is likely that strain-profiling approaches underestimated the microbial transmission rates.

**Clinical outcomes**. The primary outcome was resolution of diarrhoea 4 weeks after the end of treatments. Secondary outcomes included: resolution of diarrhoea 1, 2 and 8 weeks after the end of treatments; decrease of diarrhoea until grade G1 or lower 1, 2, 4 and 8 weeks after the end of treatment; discontinuation or reduction of treatment with TKIs.

**Randomisation and blinding**. Blocked randomisation of subjects was performed by an external individual not involved in the study. An online random number generator software (https://www.sealedenvelope.com) was used to provide random permuted blocks with a block size of four and an equal allocation ratio; the sequence was hidden until the interventions were assigned. To mask treatments to recipients, both infusate bottles and syringes were covered with dark-coloured paper before the infusion, and the patients were unable to see the endoscopic display during the procedure. Moreover, the physicians who evaluated patients at follow-up were not aware of the treatment being administered.

**Sample size calculation and statistical analysis**. To calculate sample size, we assumed a 20% resolution rate of diarrhoea in the placebo arm[16] and an 80% resolution rate of diarrhoea in the FMT arm at 4 weeks of follow-up. Using a two-tailed $p$-value of 0.05 and a power of 80% ($b = 0.20$), the enrolment of 10 patients per group was required. Sample size was calculated with an online software (https://clincalc.com/). Analyses were performed both on an intention-to-treat and per-protocol basis. Differences among groups were assessed with a two-tailed Wilcoxon rank-sum test for continuous data and with Fisher's exact probability test (using two-tailed $p$ values) for categorical data. Differences in cure percentages were determined with Fisher's exact test (with two-tailed $p$ values). The Bonferroni adjustment was applied for multiple secondary outcomes. Statistical analyses were performed through an online calculator (http://www.graphpad.com/quickcalcs/), with SPSS version 25.0 (SPSS Inc., Chicago, IL, USA), and with Microsoft Excel for Mac (Microsoft Excel. Redmond, Washington: Microsoft, 2011). For microbiome analysis, statistical differences between group means were calculated using a two-tailed Wilcoxon rank-sum test, through the R statistical software package (R Core Team, Vienna, Austria).

**Reporting summary**. Further information on research design is available in the Nature Research Reporting Summary linked to this article.

## Data availability
Nucleotide sequences are available in the Sequence Read Archive under the bioproject accession PRJNA643802. Clinical data of individual patients will be shared, after proper de-identification, upon reasonable request to the corresponding author from colleagues who want to analyse in deep our findings, from now to the next 3 years. All remaining relevant data are available in the article, supplementary information, or from the corresponding author upon reasonable request.

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

## Acknowledgements

G.C., G.I. and L.M. received grants in the field of faecal microbiota transplantation from the Italian Ministry of Health.

## Author contributions

G.C., G.I., E.R. and G.S. conceived and designed the study; G.C. and G.I. wrote the study protocol; E.R., G.S., R.I. and G.T. recruited and followed up patients; G.I., C.R.S. and G.C. screened FMT donors; L.M., G.Q. and M.S. prepared faecal infusates, stored donor and patient faecal samples, and extracted DNA for microbiome analysis; E.R. and C.R.S. built the clinical dataset; L.R.L. and G.C. performed FMT procedures; F. Armanini generated metagenomic data; A.M.T., A.B.M., F.Asnicar and N.S. analysed the metagenomic data; G.I. performed statistical analysis of clinical data; A.M.T. and N.S. performed statistical analyses of microbiome data; G.I., A.M.T., G.S., C.C., A.G., N.S. and G.C. analysed and interpreted data; G.I., A.M.T., N.S. and G.C. wrote the paper. All authors critically revised the paper for important intellectual content.

## Competing interests

A.G. reports personal fees for consultancy for Eisai S.r.l., 3PSolutions, Real Time Meeting, Fondazione Istituto Danone, Sinergie S.r.l. Board MRGE and Sanofi S.p.A, personal fees for acting as a speaker for Takeda S.p.A, AbbVie and Sandoz S.p.A and personal fees for acting on advisory boards for VSL3 and Eisai. G.C. has received personal fees for acting as advisor for Ferring Therapeutics. G.I. has received personal fees for acting as speaker for Biocodex, Danone, Metagenics, and for acting as consultant/advisor for Ferring Therapeutics, Giuliani, Metagenics. None of the other authors has any competing interest to disclose.
