## [Peer Review File · Nature Communications]

REVIEWERS' COMMENTS:

Reviewer #1 (Remarks to the Author):

I have no further comments following a review of the authors' response to reviewers' comments

Reviewer #2 (Remarks to the Author):

1. One of my primary concerns was that all patients underwent a bowel prep which puts the sham group at a disadvantage as it will decrease the diversity of an already disrupted community. That is likely why we see nearly no improvement in the sham group.

The authors have responded to this by providing a separate explanation for each of these using two arguments that 1) bowel prep does not affect gut microbiota by referencing a study that used PCR-DGGE method and 2) other groups have used water as a control.

There is sufficient data that bowel prep does indeed alter gut microbiome in studies that use next generation sequencing platforms as the one used by the authors (<https://pubmed.ncbi.nlm.nih.gov/27015015/>). The authors did not measure the microbiome before and after bowel prep in their cohort so they cannot definitively establish this. The authors then described several large trials that have used water as a sham placebo but fail to mention that none of them gave a bowel prep prior to treatment and my concern was not based on the fact that water cannot be a placebo but rather that the authors could have further disrupted an already vulnerable community with bowel prep and that can explain the lack of any response in the sham group.

The authors then argue against using autologous stool as a comparator group and I am not sure if they got confused with my comment "sham FMT refers to patients own stool". I had suggested that they should have used the patient's own stool (allogenic FMT) as a control group and all the examples provided by the authors do exactly that- use the patient's own stool as the comparator and not autologous stool as the authors mention in the rebuttal. I am not sure what the authors mean by lack of protocol for collecting stool from the patients with diarrhea. A sham/allogenic FMT would follow the exact same protocol for collection and processing as the donor and all the examples outlined by the authors in fact do this.

2. The authors did not respond to my comment

"The data is too preliminary to discuss the role of engraftment in driving improvement as the study lacks an appropriate control group. Moreover it would be helpful if the authors consider tracking the microbes longitudinally (given they have the samples) to determine if changes in relative abundance of microbes such as Akkermansia correspond to recurrence of diarrhea." Further the authors did not perform longitudinal sampling on the sham group and compare the active FMT to time point 0 of the sham group so we do not know the level of variability/changes in the microbiome in the sham group following intervention.

Overall my enthusiasm is in fact further dampened following author's response as it highlights additional drawbacks in the study design and the interpretation of the results.

Reviewer #3 (Remarks to the Author):

Authors have satisfactory assessed all my queries.

Reviewer #4 (Remarks to the Author):

All my comments have been addressed appropriately. Looks good!

We would like to thank reviewers for the careful assessment of our paper and for their precious comments, that improved the quality of our paper.

We have done our best to address comments satisfactorily, and hope that you will appreciate the revised version of the paper. Please find below a point-to-point reply to the comments from reviewers.

Best regards

Gianluca Ianiro, on behalf of all co-authors

POINT-BY-POINT REPLY TO REVIEWERS

Reviewer #1 (Remarks to the Author):

I have no further comments following a review of the authors' response to reviewers' comments

R: We are grateful for your comments and for the time you have spent in assessing our manuscript. Thank you.

Reviewer #2 (Remarks to the Author):

1. One of my primary concerns was that all patients underwent a bowel prep which puts the sham group at a disadvantage as it will decrease the diversity of an already disrupted community. That is likely why we see nearly no improvement in the sham group.

The authors have responded to this by providing a separate explanation for each of these using two arguments that 1) bowel prep does not affect gut microbiota by referencing a study that used PCR-DGGE method and 2) other groups have used water as a control.

There is sufficient data that bowel prep does indeed alter gut microbiome in studies that use next generation sequencing platforms as the one used by the authors

(<https://pubmed.ncbi.nlm.nih.gov/27015015/>). The authors did not measure the microbiome before and after bowel prep in their cohort so they cannot definitively establish this. The authors then described several large trials that have used water as a sham placebo but fail to mention that none of them gave a bowel prep prior to treatment and my concern was not based on the fact that water cannot be a placebo but rather that the authors could have further disrupted an already vulnerable community with bowel prep and that can explain the lack of any response in the sham group.

The authors then argue against using autologous stool as a comparator group and I am not sure if they got confused with my comment “sham FMT refers to patients own stool”. I had suggested that they should have used the patient’s own stool (allogenic FMT) as a control group and all the examples provided by the authors do exactly that- use the patient’s own stool as the comparator and not autologous stool as the authors mention in the rebuttal. I am not sure what the authors mean by lack of protocol for collecting stool from the patients with diarrhea. A sham/allogenic FMT would follow the exact same protocol for collection and processing as the donor and all the examples outlined by the authors in fact do this.

R: We acknowledge that bowel preparation could impair gut microbiota. However, it is mandatory when FMT is delivered through colonoscopy. Actually, among the studies that have used water as placebo, some of them (those delivering FMT by colonoscopy) have used bowel preparation before treatments (<https://pubmed.ncbi.nlm.nih.gov/32014035/>; <https://pubmed.ncbi.nlm.nih.gov/28214091/>; <https://pubmed.ncbi.nlm.nih.gov/29980607/>). It is true that we did not assess the microbiome before and after bowel preparation in our cohort, but this was not the aim of this study.

We remark that patients in both treatment arms have received bowel preparation before donor or placebo FMT, so this population was homogeneous by this point of view.

We have added the limitations raised by the reviewer in the discussion section.

2. The authors did not respond to my comment

“The data is too preliminary to discuss the role of engraftment in driving improvement as the study lacks an appropriate control group. Moreover it would be helpful if the authors consider tracking the microbes longitudinally (given they have the samples) to determine if changes in relative abundance of microbes such as Akkermansia correspond to recurrence of diarrhea.”

Further the authors did not perform longitudinal sampling on the sham group and compare the active FMT to time point 0 of the sham group so we do not know the level of variability/changes in the microbiome in the sham group following intervention.

Overall my enthusiasm is in fact further dampened following author's response as it highlights additional drawbacks in the study design and the interpretation of the results.

R: The inclusion of the placebo group in the microbiome engraftment was used as a control of the analysis (if donors shared strains with patients in the placebo arm then our methods would have been wrong). However, we acknowledge that the data is too preliminary to discuss the role of engraftment in driving clinical improvement, and this limitation has been added in the discussion section.

Reviewer #3 (Remarks to the Author):

Authors have satisfactory assessed all my queries.

R: We are grateful for your comments and for the time you have spent in assessing our manuscript. Thank you.

Reviewer #4 (Remarks to the Author):

All my comments have been addressed appropriately. Looks good!

R: We are grateful for your comments and for the time you have spent in assessing our manuscript. Thank you.